# Preparation and Swelling Behaviors of High-Strength Hemicellulose-g-Polydopamine Composite Hydrogels

**DOI:** 10.3390/ma14010186

**Published:** 2021-01-02

**Authors:** Jiayan Ge, Kaiqi Gu, Kewen Sun, Xinyue Wang, Shuangquan Yao, Xiaorong Mo, Shuilian Long, Tingting Lan, Chengrong Qin

**Affiliations:** 1Guangxi Key Laboratory of Clean Pulp & Papermaking and Pollution Control, School of Light Industrial and Food Engineering, Guangxi University, Nanning 530004, China; 1816391008@st.gxu.edu.cn (J.G.); 1805170140@st.gxu.edu.cn (K.G.); 1805170104@st.gxu.edu.cn (X.W.); 1705170126@st.gxu.edu.cn (X.M.); longshuilian77@163.com (S.L.); lantingtingd@163.com (T.L.); 2School of Computer and Information, Hefei University of Technology, Hefei 230009, China; sunkewen2014@163.com

**Keywords:** hemicellulose, polydopamine, hydrogel, mechanical property, pH-sensitive, drug release

## Abstract

Hemicellulose-based composite hydrogels were successfully prepared by adding polydopamine (PDA) microspheres as reinforcing agents. The effects of PDA microsphere size, dosage, and nitrogen content in hydrogel on the mechanical and rheological properties was studied. The compressive strength of hydrogel was increased from 0.11 to 0.30 MPa. The storage modulus G’ was increased from 7.9 to 22.0 KPa. The gaps in the hemicellulose network are filled with PDA microspheres. There is also chemical cross-linking between them. These gaps increased the density of the hydrogel network structure. It also has good water retention and pH sensitivity. The maximum cumulative release rate of methylene blue was 62.82%. The results showed that the release behavior of hydrogel was pH-responsive, which was beneficial to realizing targeted and controlling drug release.

## 1. Introduction

At present, compounds from biomass cannot be simultaneously isolated, and at least one of the polymers is damaged and discarded during the process [1]. This process leads to a huge waste of biomass resources. An effective solution is to develop methods for the whole component utilization of biomass [2,3]. It can be realized by green pretreatment technology [4] and efficient clean separation technology [5]. During pretreatment methods, hemicellulose is extracted [6]. This is the premise and core of the effective utilization of biomass. There are a series of side effects in acid or alkali pretreatment [7,8], such as cellulose degradation, lignin dissolution and low-purity hemicellulose sample. One can use extraction [9] or autohydrolysis [10] in the absence of acids or alkaline to promote cellulose and lignin recovery. Water and woody biomass are considered as reagents in autohydrolysis. Hemicellulose depolymerization is catalyzed by hydronium ions from both water and self-generated compounds (e.g., acetic, uronic and phenolic acids) [11]. Autohydrolysis technology covers a range of treatments, including both hydrothermal pretreatment [12,13] and steam-based processes [14]. Other advantages of hydrothermal pretreatment are low pollution, high efficiency, low energy consumption and high cellulose recovery [15].

There are wide applications for extracted hemicellulose. Xylose, the main degradation product of hemicellulose, is a type of food additives [16]. A novel dietary fiber containing xylose was prepared by Lv et al. [17]. Hemicellulose is also widely used in the field of bioenergy. It provides liquid fuel in a world thirsty for transportation fuel [18,19]. Lyczakowski et al. [20] reported that removal of glucuronic acid from xylan is a strategy to enhance the conversion of woody biomass to sugars for bioenergy, in which Ethanol yields were doubled compared with those obtained from non-treated material. In addition, the greatest potential is the application of hemicellulose-based polymer composites. Nanofibrils and other modified hemicellulose-based polymer composites are promising renewable alternatives for sustainable composite materials [21]. Guan et al. [22] reported that organic–inorganic composite films were prepared with different proportions of hemicellulose and clay, and the composite films have higher thermal performance and lower optical transparency. New bioremediation composite material was prepared from hemicellulose and polylactic acid for three-dimensional printing [23]; however, this composite material has strong hydrophilicity and weak mechanical strength, and the characteristics would limit its widespread applications.

In this paper, bagasse hemicellulose was extracted from hydrothermal pretreatment. Hemicellulose composite hydrogels were prepared by using polydopamine microspheres. The effects of microsphere (polydopamine, PDA) size, PDA dosage and nitrogen content on the rheological and mechanical properties of composite hydrogels were studied. The physicochemical properties of the composite hydrogel were analyzed by Fourier transform infrared spectroscopy (FTIR). The microstructure of the composite hydrogel was characterized by Scanning Electron Microscopy (SEM). The swelling properties of the composite hydrogels were analyzed, such as water retention, pH sensitivity and drug release. Composite hydrogels with high mechanical properties and swelling properties were obtained. This study provides theoretical support for the biomedical application of hemicellulose hydrogels.

## 2. Materials and Methods

### 2.1. Materials

Bagasse was provided by a local sugar mill (Hua Jin Paper Co. LTD, Guangxi, China). The chemical composition of bagasse was determined by sulfuric acid hydrolysis according to the NREL method [24]. The basic method and process were described by Ge et al. [15]. The mean value was calculated after three measurements. The contents of cellulose, hemicellulose, and lignin in bagasse were 46.88%, 20.91% and 23.41%, respectively. Dopamine was purchased from Sigma-Aldrich (St. Louis, MO, USA). Other assay reagents were obtained from Aladdin (Shanghai, China).

### 2.2. Extraction and Modification of Hemicellulose

Hemicellulose was efficiently extracted from bagasse by hydrothermal pretreatment [12]. Under the condition of a solid-liquid ratio of 1:8, sodium hydroxide was used to pre-adjust the pH of the extraction solution. The dosage of 3.9 mol·L^−1^ sodium hydroxide is 4.0%. The reactor was heated to 170 °C for 60 min. The final pH of the hydrolysate was 4. The hydrolytic solution after membrane separation was neutralized. Hemicellulose samples were obtained by centrifugal precipitation. It was analyzed by FTIR [13], high efficiency anion exchange chromatography (HPAEC) [12].

Hemicellulose was modified by ammoniation [25]. 0.5 g hemicellulose was dissolved in 25 mL pure water. It was fully dissolved at 50 °C. Then, 0.05 g ammonium persulfate was added. The amount of acrylamide added was 2.0, 3.0, 4.0, 5.0 and 6.0 g, respectively. The mixed solution reacts at 50 °C for 4 h. The reaction was terminated by adding 1.0 mL of aqueous hydroquinone solution (2.0%). The solution was dried in an oven at 50 °C. The sample was placed in a mixture of N, N-dimethyl formamide (DMF) and glacial acetic acid (v:v = 1:1). The unreacted monomers and homopolymers were removed by soaking at room temperature for 12 h. The sample was washed multiple times with acetone. The washed samples were dried at 50 °C. The modified hemicellulose was obtained. The nitrogen contents in the modified hemicellulose samples were analyzed by an elemental analyzer (FlashSmart, ThermoFisher, Waltham, MA, USA). It was 0.10, 0.23, 0.45, 0.55 and 0.80%, respectively. The above data is the average of the three measurements. The standard deviations for these values were 0.017, 0.010, 0.021, 0.025 and 0.031%, respectively. That means the experimental results are valid.

### 2.3. Preparation of PDA Microspheres

First, 0.2 g of dopamine was dissolved in 200 mL pure-water. pH of the reaction solution (7.5–9.0) was adjusted by 5.0 mol·L^−1^ NaOH. The reaction was conducted at 60°C and 500 rpm for 1–10 h. The prepared PDA microspheres were washed with ethanol and centrifuged at 10,000 rpm. Their dimensions were analyzed by transmission electron microscopy (TEM) (Hitachi HT7700, Hitachi, Tokyo, Japan) (Appendix A). It was 50, 100, 150, 200 and 250 nm, respectively. The uncertainty of the data was calculated. The results are shown in Table 1. The reasons for the uncertainty are the tightness of the instrument and the self-cohesion of PDA microsphere. Therefore, the samples were stored in ethanol solution to prevent PDA microspheres from self-aggregating.

### 2.4. Preparation of Composite Hydrogels

Modified hemicellulose (0.5 g) was added to 25 mL pure-water. The mixture was stirred at 60 °C until it was completely dissolved. Initiator ammonium persulfate (0.5 g), acrylamide (3.0 g) and cross-linking agent N, N-methylene diacrylamide (0.25 g) were in turn added and stirred to mix evenly. Then, different dimensions and dosages of PDA microspheres were added to the mixture. Complex hydrogel was prepared after 1 h. The synthesis conditions of the composite hydrogel are shown in Table 1.

### 2.5. Mechanical and Rheological Properties of Hydrogels

The compression properties of hydrogel was measured using an electronic universal material testing machine (Instron Inc., Boston, MA, USA) [26]. The sample was compressed at a rate of 3 mm·min^−1^ until the compression rate reached 90%. The rheological properties of hydrogel were measured by a rheometer (R/s-cc, Brookfield, Middleboro, MA, USA) [27]. All data were averaged after three measurements and the errors were calculated.

The functional groups of hydrogel were analyzed by FTIR (Nexus 470, HP, Palo Alto, CA, USA). The surface morphology of hydrogel was analyzed by SEM (S-3400 N, Hitachi, Tokyo, Japan).

### 2.6. Evaluation of Biomedical Application Value

Hydrogel was placed in distilled water to achieve adsorption equilibrium. The water retention (Wr) properties of hydrogel was determined [28]. The adsorption equilibrium hydrogel was placed in an oven at 60 °C for the dehydration experiment. The real-time quality of the hydrogel was calculated every 2 h.

The biomedical application of hydrogel is directly influenced by pH sensitivity and drug release. The pH sensitivity of hydrogels was analyzed [29]. A certain amount of dry hydrogel sample was immersed in deionized water with different pH values at room temperature. The swelling hydrogel was removed after a certain period of time. The weight gain of hydrogel was calculated. The swelling rate of hydrogels was calculated according to Equation (1):SR = (M_2_ − M_1_)/M_1_(1)
where SR is the swelling rate of hydrogels (%). M_1_ is the weight of the dry hydrogel sample (%). M_2_ is the weight of hydrogel sample after water absorption (%).

Methylene blue was used as a sustained release drug model in hydrogels. First, the methylene blue adsorption experiment was carried out. The hydrogel (0.10 g) was fully swollen in PBS buffer solution at 25 °C. It was added to a methylene blue solution of 50 mg·L^−1^ for 24 h. The absorbance of methylene blue in the remaining solution was measured by a UV spectrometer. Combined with the standard curve of methylene blue solution, the concentration of methylene blue in the remaining solution was obtained. The adsorption capacity of hydrogel was calculated [30].

The adsorbed saturated hydrogel was placed in PBS buffer solution at 25 °C for the release study. A small amount of buffer was taken at regular intervals for methylene blue concentration determination. The PBS buffer solution in the release system was supplemented in a timely manner. The cumulative release rate was calculated [31].

The above data of swelling property and loading property are the mean values after three measurements, and the errors of the data are calculated. It is used to evaluate the validity of the results.

## 3. Results and Discussion

### 3.1. Characterization of Hemicellulose Samples Extracted by Hydrothermal Pretreatment

The high value utilization of hemicellulose is influenced by the yield and purity. In fact, the extraction of hemicellulose by hydrothermal pretreatment was accompanied by the dissolution of lignin. Therefore, the characteristics of hemicellulose sample were analyzed. The results are shown in Figure 1.

Figure 1a shows FTIR of hemicellulose sample. There was a strong absorption peak at 3421 cm^−1^. This is due to the O–H stretching vibration of polar groups connected by intermolecular and intermolecular hydrogen bonds [32]. The C–O–C stretching vibration peak exists at 1045 cm^−1^. They are characteristic absorption peaks of hemicellulose. The β-glycosidic bond between the sugar units in hemicellulose was reflected in the absorption peak at 899 cm^−1^ [33]. The characteristic absorption peak of arabinoxylan was 990 cm^−1^. The absorption peak generated by the aromatic skeleton vibration was at 1610 cm^−1^. This is the characteristic peak of residual lignin in hemicellulose sample [34]. The results showed that the purity of hemicellulose was higher. It contains a small amount of lignin. The relative content of lignin in the sample was 10.88%.

The extraction rate of hemicellulose was 43.45%. The chemical composition was analyzed by HPAEC. It was mainly composed of xylose, glucose, arabinose and galactose. The contents were 65.24%, 6.45%, 3.81% and 1.90%, respectively. It means that the extraction hemicellulose has a good application prospect.

### 3.2. Effect of PDA Particle Size on The Mechanical and Rheological Properties of Hydrogel

PDA surface modification is increasingly widely used in the surface modification of biomaterials. This is due to its convenience, simplicity and good biocompatibility [35]. PDA can provide active functional groups for the secondary reaction on the surface of materials, which is conducive to the anchoring of functional molecules or functional groups and obtains biological activity. However, PDA molecules have self-polymerization behavior, which is mainly affected by the size and amount of PDA microspheres. Therefore, the effect of the particle size of PDA microspheres on the compressive strength and rheological properties of composite hydrogel was studied. The particle size of the PDA microsphere was 50, 100, 150, 200 and 250 nm, respectively. The nitrogen content of hemicellulose was 0.45%, and the addition of PDA microspheres was 1.28%. The results are shown in Figure 2.

The compressive strain of pure hemicellulose hydrogels and the composite hydrogels with PDA microspheres of different sizes were compared and analyzed. The error values of the data after three measurements were 4.69, 7.01, 5.85, 5.37, 5.07 and 4.89 MPa, respectively. That means the results are valid. Figure 2a shows that the compressive strength of the composite hydrogel reached the maximum value of 0.19 MPa when the PDA microsphere size was 50 nm. Although the compression strength of the other composite hydrogel was lower than that of the PDA microsphere size at 50 nm, it was higher than that of conventional hemicellulose hydrogels (0.11 MPa). This result indicates that the addition of PDA microspheres is beneficial to improve the compressive strength of composite hydrogel. On the other hand, the compressive strength of the composite hydrogel decreases with increasing PDA microsphere size. PDA contains abundant hydroxyl groups of catechol and active amino groups [36], which penetrate into the hemicellulose cross network after a secondary reaction with ammoniated hemicellulose. As a result, the hydrophobic aggregation of the chain segments inside the gel was blocked. The phase separation inside the gel was inhibited. Finally, the gel strength was improved significantly. When the size of the PDA microspheres continues to increase, the PDA microspheres with larger sizes have swelling rates that are smaller than those of the surrounding gel. Therefore, the location of these PDA microspheres becomes a stress concentration point during the swelling process of hydrogel, which leads to cracks inside the gel and ultimately leads to a significant decrease in the compressive strength of hydrogel.

Figure 2b shows the change curve of storage modulus (G’) and loss modulus (G”) of hydrogel with frequency (0.1–10 Hz) under the condition of 1% strain and different PDA microsphere sizes. Chemical cross-linked hydrogels have stable mechanical properties [37]. Usually, G’ is much larger than G” in the frequency range. It shows that the elastic response was stronger than the viscous response. G’ hardly changed within 10 Hz. It indicates that the network structure of hemicellulose-based hydrogels was well formed. For the composite hydrogels with PDA microspheres added, the increase in G’ value was more obvious over the whole frequency range. Although the rheological results of composite hydrogel may be affected by lower water content, the main influencing factor is the formation of a compound network based on hemicellulose and PDA microspheres. G’ of hydrogel increased from 7.9 [38] to 20.0 KPa (PDA microsphere size 50 nm) with PDA microsphere addition. This is due to the enhanced effect of PDA microspheres in the hemicellulose network. Therefore, the optimal PDA microsphere size of 50 nm was obtained.

### 3.3. Effect of PDA Microsphere Dose on the Mechanical and Rheological Properties of Hydrogel

The self-aggregation behavior of PDA microspheres was promoted with the increased dosage of PDA microsphere [39]. The appropriate dosage of PDA microspheres is important for improving the performance of composite hydrogel. The effects of PDA microsphere dosage on the mechanical and rheological properties of hydrogel were studied. It was 0.64%, 1.28%, 1.92%, 2.56% and 3.20%, respectively. The nitrogen content was 0.45%, and the particle size of PDA microspheres was 50 nm. The results are shown in Figure 3.

The effect of PDA microsphere dosage on the compressive strength of composite hydrogel is shown in Figure 3a. When the dosage of PDA microspheres was lower than 1.92%, the compression stress of composite hydrogel increased from 0.11 to 0.19 MPa. It was remained unchanged with the dosage increase to 2.56%. However, the compressive strength of composite hydrogel decreased significantly with the dosage increased. The main reason is that the more active sites of PDA are produced with the increase dosage of PDA microspheres. Its binding to ammoniated hemicellulose was enhanced. The density of the network chain inside the composite hydrogel is increased. The interaction force between the segments was strengthened. The network skeleton of hydrogel was more rigid. Thus, the compressive strength of hydrogel was increased [40]. However, its self-aggregation behavior was enhanced with the increase of PDA microsphere addition. The uniform distribution of PDA microspheres inside the hydrogel was destructed. The “independent region” was appeared. The compressive strength was significantly decreased. The error values were 4.69, 4.84, 6.23, 7.13, 7.79 and 4.65 MPa, respectively. The result is valid as the error is within a reasonable range.

The frequency changes in G’ and G” of hydrogel with different PDA microsphere additions are shown in Figure 3b. G’ increased from 7.9 to 27.0 KPa. The maximum G’ was obtained at a PDA microsphere dosage 2.56%. However, the integrity of the gels was destroyed with the addition of excessive PDA microspheres (3.20%). The rheological properties of hydrogel showed a decrease in G’. G” varies from 0.3 to 1.7 KPa. This result indicates that the viscoelasticity of the hydrogel decreases gradually with the addition of PDA microspheres. This is due to the strong stability and water insolubility of PDA, which restricts the viscosity flow of polymers such as hemicellulose inside hydrogel [41]. Moreover, the self-aggregation behavior after the addition of excessive PDA microspheres was affected. The results showed that the optimal dose of PDA microspheres was 2.56%.

### 3.4. Effect of Nitrogen Content on Mechanical and Rheological Properties of Hydrogel

The performance of composite hydrogel was improved by the ammoniation modification of hemicelluloses [42]. The ammoniation quantification of hemicellulose was mainly expressed by the nitrogen content in hemicellulose. The effects of hemicellulose on the compressive strength and rheological properties of hydrogel were studied by adjusting the nitrogen content of hemicellulose. The size of PDA microspheres was 50 nm, and the addition amount was 2.56%. The results are shown in Figure 4.

The results show that the error corresponding to the experimental data varies between 3.99 and 11.32 MPa. The effectiveness of the experiment was confirmed. Figure 4a shows the changes in the compression and rheological properties of the hydrogels with different nitrogen contents of hemicellulose. Unmodified hemicellulose hydrogel has a smaller compressive stress. However, the compressive strength of the composite hydrogel was increased with the increase of nitrogen content. The maximum compressive strength of composite hydrogel was 0.30 MPa at the hemicellulose nitrogen content 0.80%. The short chain of hemicellulose formed by gel network fracture has a certain crosslinking effect. Large-scale slippage is difficult to be formed and network skeleton is difficult to be destroyed. The regularity of the molecular network was maintained by the viscous flow of the linear polymer. However, a lot of energy was consumed in the process [43]. The hemicellulose network space was filled by PDA microspheres. Furthermore, the chemical crosslinking between them was formed by the addition of ammoniated hemicellulose. Chemical crosslinking was enhanced with the increase of nitrogen content in hemicellulose. The skeleton structure of the hydrogel network was strengthened. The compressive strength and storage modulus G’ of the composite hydrogel were significantly improved (Figure 4b). Due to the physical filling and chemical cross-linking of PDA microspheres in the hemicellulose network, the viscoelasticity inside hydrogel was inhibited. G” of the composite hydrogel was increased with increasing nitrogen content in hemicellulose. Therefore, the optimal nitrogen content of hemicellulose was 0.80%.

In conclusion, the optimal conditions were as follows: the PDA microsphere size was 50 nm, the dosage was 2.56%, and the nitrogen content of ammoniated hemicellulose was 0.80%. The compressive strength of the composite hydrogel was increased from 0.11 to 0.30 MPa. The storage modulus G’ was increased from 7.9 to 22.0 KPa. It means that the composite hydrogel has excellent compressive and rheological properties. In addition, the morphology and physicochemical properties of the hydrogel with and without the addition of PDA microspheres were analyzed. The change in the mechanical properties of composite hydrogels was verified.

### 3.5. Morphology and Physicochemical Properties of Hydrogel

The composite hydrogel and the traditional hemicellulose hydrogel were prepared separately. The morphology of hydrogels was analyzed. The binding of PDA in hydrogel was examined by FTIR and SEM. The mechanism for improving the mechanical properties of composite hydrogel was analyzed. The results are shown in Figure 5.

Hemicellulose extracted by the hydrothermal method contains a small amount of lignin, so the final hydrogel sample is presented as reddish brown [44]. PDA microspheres are black particles, which are uniformly distributed in the hemicellulose network. The preparation process of the composite hydrogel is shown in Figure 5a. It mainly includes ammonization modification of hemicellulose, chemical crosslinking between hemicelluloses, PDA addition and gel network formation. Figure 5b shows the changes in the main functional groups in composite hydrogel. The hydrogels grafted to acrylamide by hemicellulose modification showed obvious absorption peaks at 1676 cm^−1^ and 1332 cm^−1^, which were generated by C=O and C–N stretching vibrations in acrylamide [45]. A new absorption peak appeared at 1543 cm^−1^ with adding PDA. This is due to the stretching vibration of the C=N, N=O groups and the skeleton vibration of the benzene ring, which is generated by the reaction of the PDA with the amide group in polyacrylamide [46]. It means that the addition of PDA provides a large number of active sites binding to the amide group in the hemicellulose molecular network, and the chemical crosslinking inside the gel is strengthened. Figure 5c,d show the microscopic morphology of hydrogel. Compared with the former loose and porous network structure, the network distribution of composite hydrogels was more compact. This is due to the gaps between hemicellulose networks were filled by PDA. It provides a large number of active sites for the binding of amide groups in hemicelluloses. The entanglement of hemicellulose molecular chains was promoted. The mechanical properties of hydrogel were strengthened.

### 3.6. Swelling and Adsorption Properties of Hydrogels

At present, temperature-sensitive and pH-responsive intelligent hydrogels have been studied most; pH-responsive hydrogels have attracted increased attention [47]. The pH-responsive hydrogel entrapped the drug molecules in the hydrogel network and enabled the hydrogel carrier to release the targeted drug in response to the changes of the body’s pH. The PDA microspheres filled the gap between the hemicellulose network and cross-linking, which improved the mechanical properties of the hydrogel. However, PDA is highly water-insoluble and has many active sites. The effects of hydrogel on water retention, swelling rate, drug loading and drug release rate were studied. The results are shown in Figure 6. The errors of the experimental data are calculated. They are all within reasonable bounds.

The water retention rate of the hydrogel at a specified time is shown in Figure 6a. With the extension of time, the water retention rate of traditional hydrogels decreased from 94.30% to 53.82%. This was due to the hemicellulose hydrogel has a three-dimensional network structure of polymer after water swelling. The liquid and hemicellulose molecular networks have affinity. The liquid was trapped in a network of hemicelluloses its fluidity. The internal water does not come out easily. The gel has good water retention performance [40,48]. In fact, the water trapped in the gel was due to physical and chemical adsorption. The gaps between the hemicellulose networks were filled with PDA microspheres addition. The physical adsorption was reduced. However, a large number of active sites on PDA forming chemical bonds with hemicellulose molecules, which promotes the entangling effect. The chemical adsorption of composite hydrogels was enhanced. Finally, the water retention performance of composite hydrogels decreased slightly. Within 16 h, its water retention rate decreased from 94.42% to 51.49%. It means that the composite hydrogel has good water retention properties.

The responsiveness of environment sensitive hydrogel is mainly reflected in hydrogel swelling ratio when the external environment changes. Therefore, the swelling property of hydrogel directly affects the environmental sensitivity of hydrogel. Figure 6b shows the change in the swelling rate of the hydrogel with pH. The swelling rate of traditional hemicellulose hydrogel increased from 47.51% to 50.45%. This result indicates that the pore size of the hydrogel increases with increasing pH. This was due to the strong hydrogen bonding effect in hydrogel at low pH values (pH < 4), which is not conducive to the swelling of the hydrogel. The hydrogen bonding effect of the hydrogel network was weakened with increasing pH. The swelling rate was increased. The carboxyl group was formed from the amide group in an alkaline environment. The repulsive force between hydrogel networks was enhanced. The swelling rate of hydrogel was increased. Therefore, the hydrogen bond in the hydrogel was weakened with increasing pH. The electrostatic repulsion was increased. The pore size of the hydrogel was increased. The swelling rate of the hydrogel was increased [49]. The compactness of the hydrogel network structure was enhanced by the addition of PDA microspheres (Figure 5c,d). The hardness was relatively high. The swelling degree of the composite hydrogel was decreased. It increased from 46.43% to 49.15%. The results showed that the swelling rate of the composite hydrogel was only approximately 1% lower than that of the traditional hemicellulose hydrogel. It means that the composite hydrogel has better pH sensitivity and strength and a higher swelling degree. This approach provides a new method for the preparation of intelligent hydrogel.

The adsorption capacity of traditional hemicellulose hydrogel and composite hydrogel to methylene blue was 0.35 and 0.41 mg·g^−1^, respectively. This is owing to PDA provides a large number of active sites. The results show that the strength of hydrogels with PDA was high. Then, the drug release properties of hydrogel were studied. The release behavior of drug-loaded hydrogels in simulated intestinal fluid at 37 °C (pH 7.4) was examined in Figure 6c. The drug release rate of the composite hydrogel was significantly higher than that of the traditional hemicellulose hydrogel. The drug release diagram is shown in Figure 6d. The release of drug in hydrogel was stable after sustained for 6 h. The cumulative drug release of composite hydrogel and traditional hemicellulose hydrogel was 62.82% and 47.77%, respectively. In addition, the release of the two types of hydrogel were 31.44% and 29.97% within 2 h, respectively, while the release rate of the general drug carriers under the intestinal fluid environment could reach 50%. The hemicellulose-based hydrogel has an obvious drug release effect, and the release behavior of the composite hydrogel shows more obvious pH responsiveness. This result suggests that complex hydrogels, delivered orally to the stomach as drug carriers, can reduce swelling of the gels and inhibit drug release under acidic conditions. When entering the weakly alkaline intestinal fluid, the carboxylic acid in the hydrogel can be ionized into carboxylic acid ions. Electrostatic repulsion between charges causes the large molecular chains in hydrogel was stretched. The swelling rate of hydrogel was increased. The gaps were increased. Therefore, the release rate of drug was obviously increased. The responsiveness of the pH-sensitive composite hydrogel can adjust the release and reduce the side effects of drugs in the stomach and enhance release in the intestine to achieve controlled drug release. In addition, the mechanical properties of the hydrogel were reduced after the adsorption and release of methylene blue, which is due to the partial breakdown of chemical bonds. However, the mechanical strength of composite hydrogels decreased less. This result is owed to the fact that the connections between PDA and the hemicellulose molecular chain were strengthened as the chemical bonds among hemicellulose molecules were broken. The detailed mechanism should be discussed in depth in the future research.

## 4. Conclusions

Hemicellulose-based composite hydrogels with PDA microspheres as the reinforcing phase were prepared. The optimal size of PDA microspheres was 50 nm, the addition amount of PDA microspheres was 2.56% and the nitrogen content of ammoniated hemicellulose was 0.80%. The gap of the hemicellulose network is filled by PDA, and the active site binds to the hemicellulose molecule, which causes the network to be entangled. Compared with the traditional hemicellulose hydrogels, composite hydrogels have excellent compression and rheological properties, good water retention, pH sensitivity and obvious drug release. The performance of hemicellulose hydrogel was greatly improved. This study provides theoretical support for the biomedical application of hemicellulose polymers.

## Figures and Tables

**Figure 1 materials-14-00186-f001:**
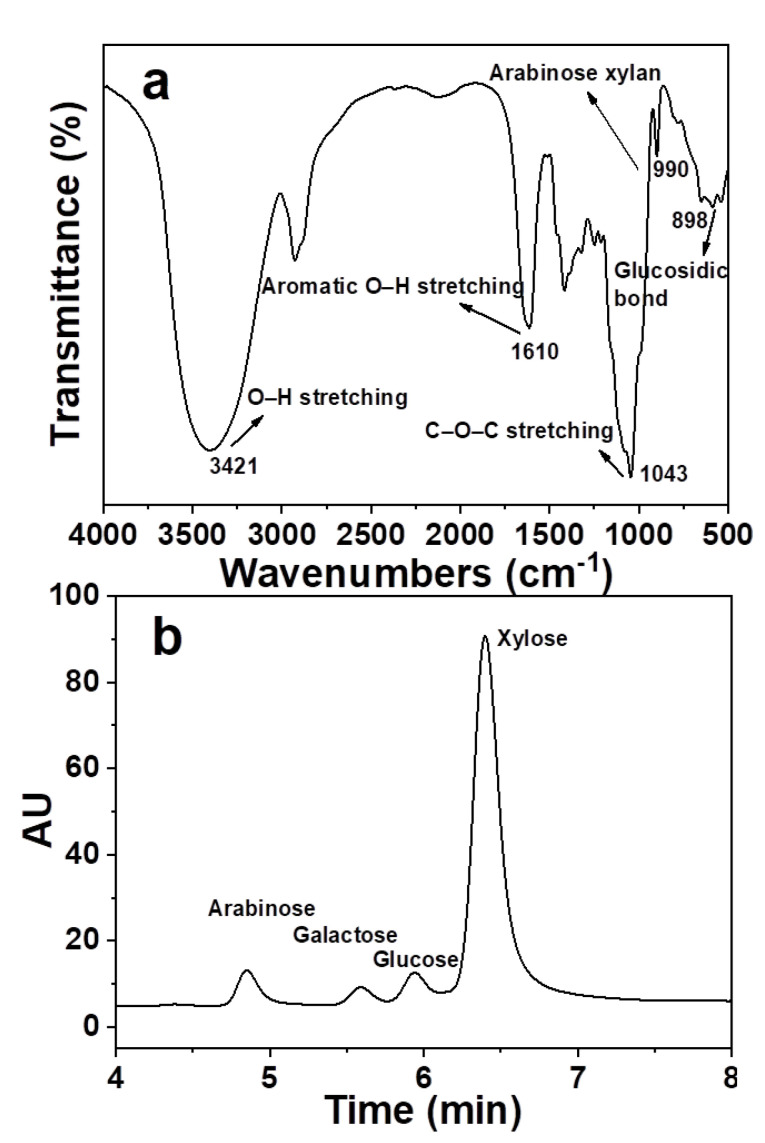
Characterization of hemicellulose sample (**a**) FTIR of hemicellulose sample and (**b**) HPAEC of hemicellulose sample.

**Figure 2 materials-14-00186-f002:**
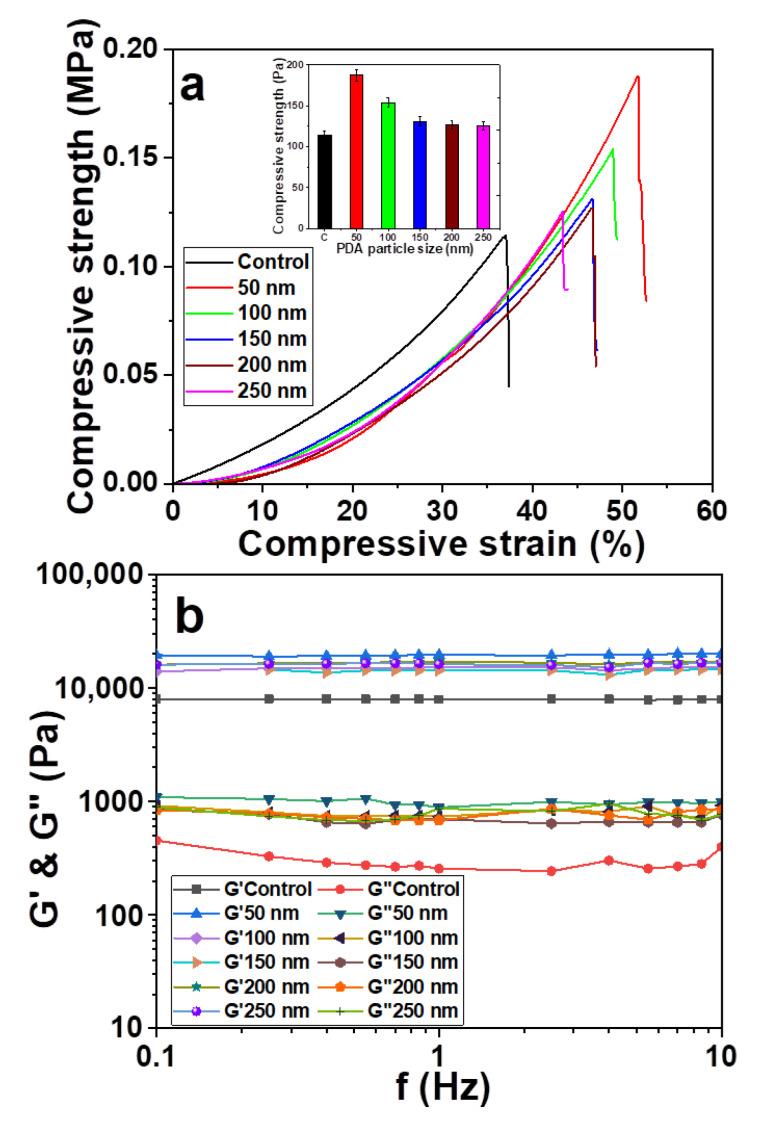
Effect of polydopamine (PDA) particle size on the (**a**) compression and (**b**) rheological properties of hydrogel.

**Figure 3 materials-14-00186-f003:**
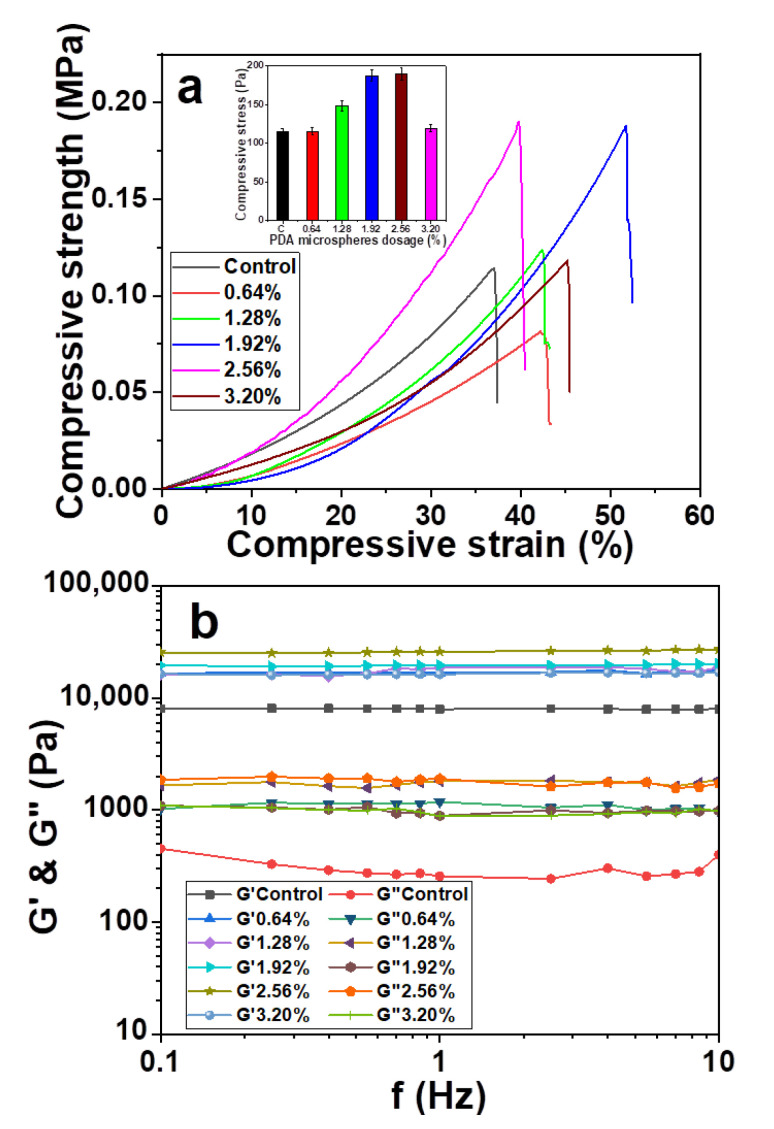
Effect of PDA microsphere dose on the (**a**) compression and (**b**) rheological properties of hydrogel.

**Figure 4 materials-14-00186-f004:**
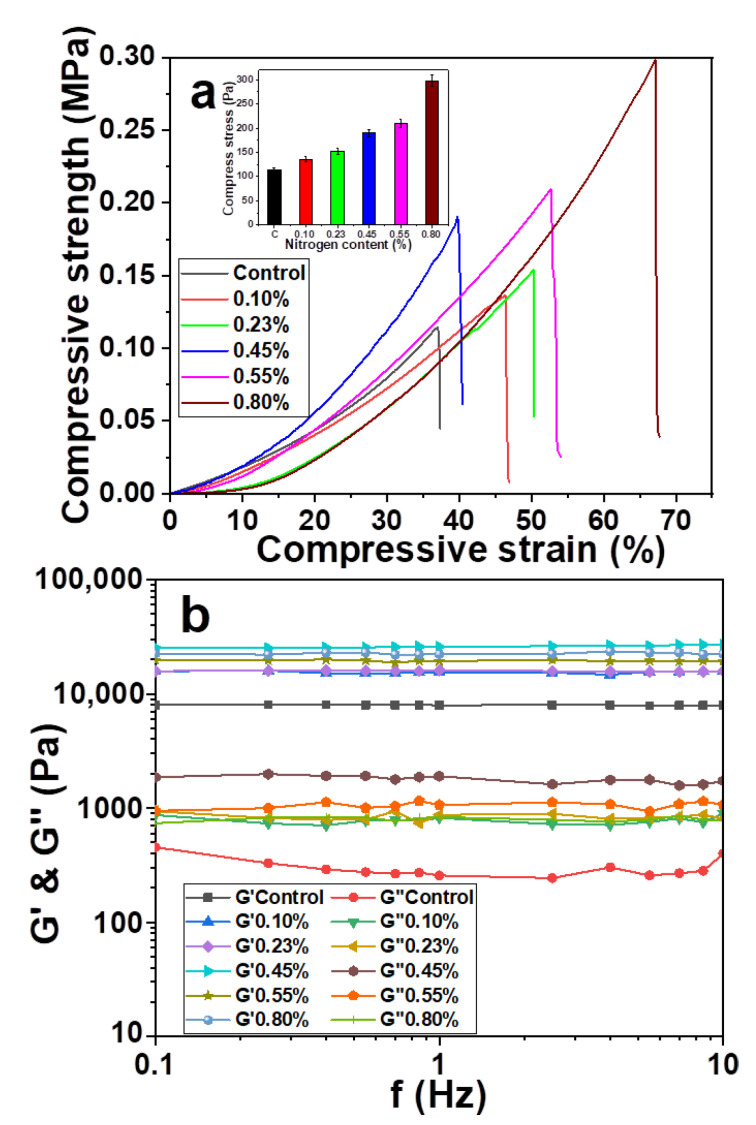
Effect of nitrogen content on the (**a**) compression and (**b**) rheological properties of hydrogel.

**Figure 5 materials-14-00186-f005:**
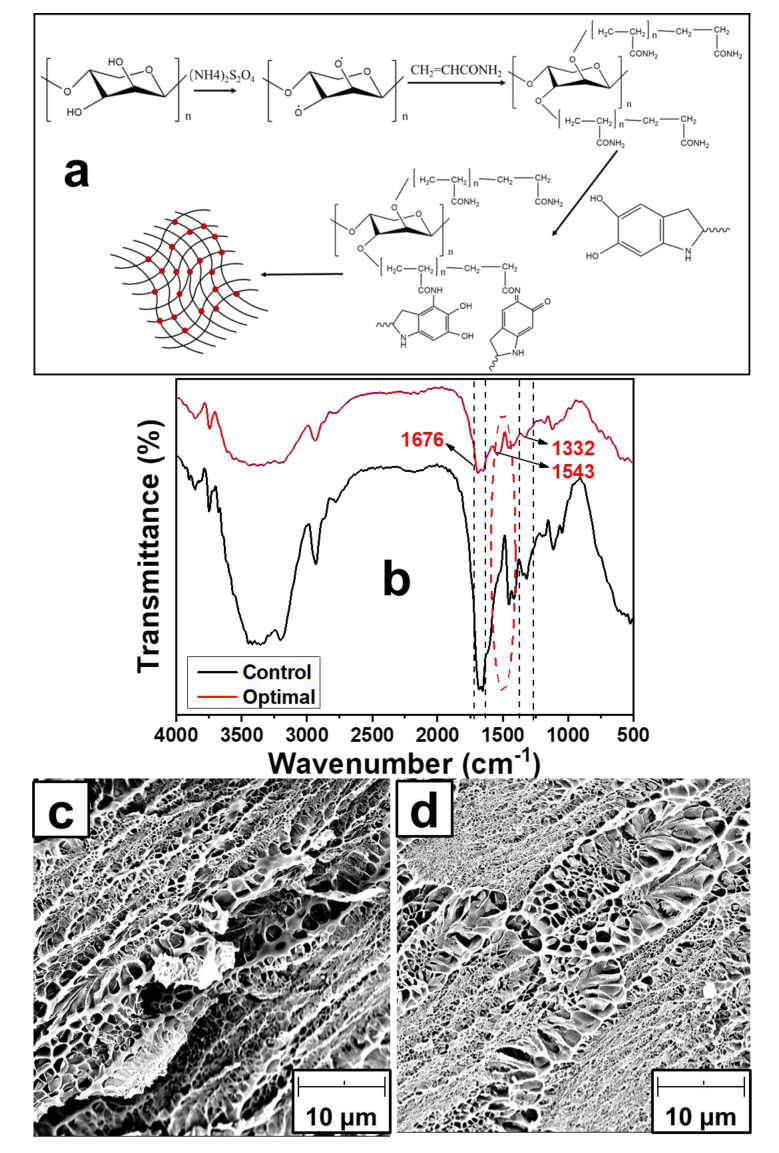
Morphology and physicochemical characteristics of hemicellulose hydrogel with and without PDA addition (**a**), Preparation process of the composite hydrogel; (**b**) FTIR of hydrogel; (**c**) SEM of traditional hemicellulose hydrogel; (**d**) SEM of composite hydrogel.

**Figure 6 materials-14-00186-f006:**
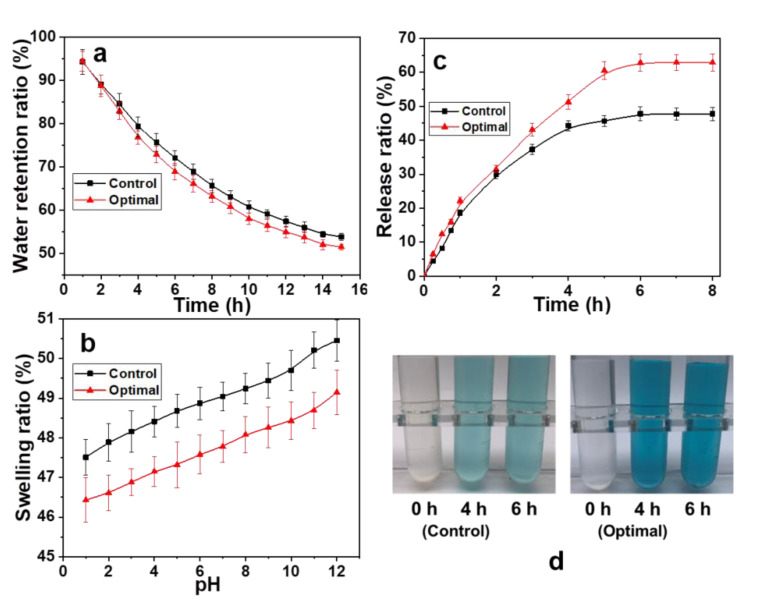
Swelling and release properties of hemicellulose hydrogel with and without PDA addition (**a**) water retention of hydrogels; (**b**) pH sensitivity of hydrogels; (**c**) drug release properties of hydrogels; (**d**) demonstration of methylene blue release in hydrogel.

**Table 1 materials-14-00186-t001:** Synthetic conditions of composite hydrogel.

Nitrogen Content (%)	PDA Microspheres Dosage (%)	PDA Microsphere Size/Uncertainties (nm)
0.10	0.64	50/9
0.23	1.28	100/11
0.45	1.92	150/7
0.55	2.56	200/12
0.80	3.20	250/9

## Data Availability

Data sharing is not applicable to this article.

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
