# Peer review of "Preparation and Swelling Behaviors of High-Strength Hemicellulose-g-Polydopamine Composite Hydrogels"

_materials, 2021, doi:10.3390/ma14010186_

Round 1

Reviewer 1 Report

Revision of Manuscript ID materials-1053206, for Materials.

Title: “Preparation and swelling behaviors of high-strength hemicellulose-g-polydopamine composite hydrogels” by Jiayan Ge et al.

In this manuscript the authors studied the mechanical and rheological properties of hemicellulose-based composite hydrogels loaded polydopamine (PDA) microspheres as reinforcing agents.

The paper is well-written, the English is sufficiently good and I think that this work could be of interest for the field of drug-delivery and materials science. The paper is technically sound, but it is limited and specific sound. The work is well structured and the proposed goals were achieved.

The manuscript contains new and preliminary information to justify publication. The methods described comprehensively. The interpretations and conclusions justified by the results. However, the manuscript should be improved and it will be worth for publication only after minor revisions as recommended below.

Major issue

I believe this manuscript is lacking in the discussion of errors and uncertainties. Authors should add information on the reproducibility of the measurements. The article could be enriched with summary tables.

Minor issues

Line 88 and Line 95 – The authors use “ultrapure-water” and “pure-water”. Is there any difference? Clarify this aspect.

PDA Microsphere size - Throughout the manuscript the authors indicate the size. It seems these are error free. This is not possible. Authors must estimate uncertainties.

Line 116 “swelling” - The Authors should better specify the procedure used for the swelling calculation.

All Figures – All figures can be enlarged. I suggest organizing in 2 rows and one column. Furthermore, all the figures must be improved in terms of font-size.

In all part of the main text, different minor typo corrections that should be performed.

Author Response

Major issue

Point 1: I believe this manuscript is lacking in the discussion of errors and uncertainties. Authors should add information on the reproducibility of the measurements. The article could be enriched with summary tables.

Response 1: Thank you very much for your valuable comments. The discussion of errors and uncertainties was added in the revised version.

Minor issues

Point 2: Line 88 and Line 95 – The authors use “ultrapure-water” and “pure-water”. Is there any difference? Clarify this aspect.

Response 2: Pure-water was used during the experiment. It has been modified in revised version.

Point 3: PDA Microsphere size - Throughout the manuscript the authors indicate the size. It seems these are error free. This is not possible. Authors must estimate uncertainties.

Response 3: The uncertainties of PDA microsphere size were evaluated. The results are shown in Table 1.

Point 4: Line 116 “swelling” - The Authors should better specify the procedure used for the swelling calculation.

Response 4: The calculation of swelling properties of hydrogels was improved in revised version.

Point 5: All Figures – All figures can be enlarged. I suggest organizing in 2 rows and one column. Furthermore, all the figures must be improved in terms of font-size.

Response 5: All Figures have been modified and rearranged in revised version.

Point 6: In all part of the main text, different minor typo corrections that should be performed.

Response 6: The different minor typo corrections were modified in revised version.

As a whole, issues the referee suggested are very pertinent, which are very helpful to modify my entire paper and thank you very much again.

Reviewer 2 Report

The manuscript submitted to Materials entitled " Preparation and swelling behaviours of high-strength hemicellulose-g-polydopamine composite hydrogels" by Jiayan Ge et al. presents the preparation and mechanical characterization of hemicellulose-g-polydopamine composite hydrogels and their biomedical evaluation with adsorption and release of methylene blue.

The overall subject has currently a growing interest and is a highly relevant subject. The overall organization of the manuscript seems logical and the rationale behind the preparation of the hydrogel composites seems logical and, in most cases, well supported by the results, however, a few inconsistencies should be addressed by the authors:

a) The introduction is confusing, and it does not highlight the subject properly. It should be clarified why these composites are a subject of interest, for the average reader will a reduced comprehension of the subject. Furthermore, there some typos and these should be proofread. Also, in lines 43, 52 and 72, et al. or co-workers should be added after the author name referenced (g. Line 43: Lu et. al.).

b) In the materials and methods section the authors describe the preparation of hemicellulose samples with different nitrogen contents (0.10, 0.23, 0.45 0.55 and 0.80%), however, the changes made in the procedure to obtain these percentages are not discriminated. The same happens in the of different PDA microsphere sizes. The procedures should report the exact quantities used by the authors to obtain all the materials used.

c) The scheme in figure 5 should be modified because the reactants above the arrows seem to be attached to the polymers. Furthermore, the scheme size should be bigger.

d) The reasons provided for the authors for the water retention differences between the control and the hydrogel composite are confusing and should be clarified (lines 308 to 316).

e) Does the hydrogel composite retain the mechanical properties after the adsorption and release studies of methylene blue? This subject is particularly important when changing the pH, since pH changes are normally associated with the supramolecular interactions of the hydrogels and therefore might change their properties.

Author Response

Point 1: The introduction is confusing, and it does not highlight the subject properly. It should be clarified why these composites are a subject of interest, for the average reader will a reduced comprehension of the subject. Furthermore, there some typos and these should be proofread. Also, in lines 43, 52 and 72, et al. or co-workers should be added after the author name referenced (g. Line 43: Lu et. al.).

Response 1: Thank you very much for your valuable comments. The novelty factor of this contribution was added in the introduction section. Some typos have been corrected in revised version.

Point 2: In the materials and methods section the authors describe the preparation of hemicellulose samples with different nitrogen contents (0.10, 0.23, 0.45, 0.55 and 0.80%), however, the changes made in the procedure to obtain these percentages are not discriminated. The same happens in the of different PDA microsphere sizes. The procedures should report the exact quantities used by the authors to obtain all the materials used.

Response 2: The details of the preparation of modified hemicellulose were added.

Point 3: The scheme in figure 5 should be modified because the reactants above the arrows seem to be attached to the polymers. Furthermore, the scheme size should be bigger.

Response 3: All Figures have been modified and rearranged in revised version.

Point 4: The reasons provided for the authors for the water retention differences between the control and the hydrogel composite are confusing and should be clarified (lines 308 to 316).

Response 4: The comparative analysis of the water retention performance of the control and the composite hydrogel was improved.

Point 5: Does the hydrogel composite retain the mechanical properties after the adsorption and release studies of methylene blue? This subject is particularly important when changing the pH, since pH changes are normally associated with the supramolecular interactions of the hydrogels and therefore might change their properties.

Response 5: The mechanical properties of the hydrogel were reduced after its adsorption and release of methylene blue. This is due to the partial breakdown of chemical bonds during adsorption and release. However, the mechanical strength of composite hydrogels decreased less. This is owed to the fact that the connection between PDA and hemicellulose molecular chain was strengthened as the chemical bonds between hemicellulose molecules are broken. The detailed mechanism will be analysed in the following research.

As a whole, issues the referee suggested are very pertinent, which are very helpful to modify my entire paper and thank you very much again.

Reviewer 3 Report

In this work by Ge and co-workers, the authors synthesized and analyzed hemicellulose-g-polydopamine composite hydrogels. These materials showed good mechanical and rheological properties as well as promising water retention capabilities. However, several issues must be addressed before this paper can be published in Materials. Please find the suggestions below:

1) Please specify more clearly the novelty factor of this contribution in the introduction section. It should be directly visible what is that separates this paper from state of the art.
2) "scanning electron microscopy (SEM) (Line 64), "transmission electron microscopy (TEM)" (Line 91) - should be capitalized
3) Experimental details were reported with insufficient rigor, which, in the current state, makes the study not reproducible by others. The ability to repeat experiments from a paper by other researchers is a requirement for scientific publication.
- Description of the extraction process, which is key for this contribution, lacks the amount of NaOH employed to adjust the pH (Section 2.2.)
- No details were provided regarding characterization "The hemicellulose samples were analyzed by FTIR, ion chromatography (IC) and gel permeation chromatography (GPC)." (Lines 80-81). (Section 2.2.)
- "Hemicellulose is modified by ammoniation [25]. The organic polymer coating is formed from ammoniated hemicellulose polymerized with dopamine. The functionality of hemicellulose composite hydrogels was further improved." (Lines 82-84) - it is not specified what the conditions and parameters of this process are. (Section 2.2.)
- "Hemicellulose (0.5 g) was added to 25 mL of pure water. The mixture was stirred at 60°C until it was completely dissolved. Initiator (NH4)2S2O8, acrylamide and cross-linking agent N, N-methylene diacrylamide were added in turn and stirred to mix evenly. " (Lines 95-97) - what are the exact amounts? (Section 2.4.)
Please correct these mistakes and other shortcomings of the experimental section, which I did not list.
4) "The hydrolytic solution after membrane separation was neutralized to neutral." (Line 79) - neutralized means already that the pH is neutral
5) Headlines should not be separated from the sections they describe (Line 129)
6) Insets in Fig. 2a, 3a, 4a are too small to read. The same is valid for Fig. 5c. Same micrographs should be enlarged and have professional scale bar markers. Currently, no conclusions can be drawn from these images as they are too tiny.
7) "The nitrogen contents in the modified hemicellulose 85 samples were analyzed by an elemental analyzer and found to be 0.10%, 0.23%, 0.45%, 0.55% and 86 0.80%." (Lines 84-86) and "Their dimensions were analyzed by transmission electron microscopy (TEM) and found to be 50 nm, 100 nm, 150 nm, 200 nm and 250 nm. " (Lines 91-92). It is puzzling how the authors obtained such round values. Please comment on that. Also, provide standard deviations for these values as there was definitely some uncertainty. I also invite the authors to include these TEM images in the final work.
8) The conclusions section should be extended to include the impact of this contribution and future outlook.

Author Response

Point 1: Please specify more clearly the novelty factor of this contribution in the introduction section. It should be directly visible what is that separates this paper from state of the art.

Response 1: The novelty factor of this contribution was added in the introduction section.

Point 2: "scanning electron microscopy (SEM) (Line 64), "transmission electron microscopy (TEM)" (Line 91) - should be capitalized

Response 2: It has been modified according to your suggestion.

Point 3: Description of the extraction process, which is key for this contribution, lacks the amount of NaOH employed to adjust the pH (Section 2.2.)

Response 3: The amount of sodium hydroxide solution was added in revised version.

Point 4: No details were provided regarding characterization "The hemicellulose samples were analyzed by FTIR, ion chromatography (IC) and gel permeation chromatography (GPC)." (Lines 80-81). (Section 2.2.)

Response 4: Relevant references were added. The details of the test analysis are described in the references.

Point 5: "Hemicellulose is modified by ammoniation [25]. The organic polymer coating is formed from ammoniated hemicellulose polymerized with dopamine. The functionality of hemicellulose composite hydrogels was further improved." (Lines 82-84) - it is not specified what the conditions and parameters of this process are. (Section 2.2.)

Response 5: The conditions and parameters of this process were added in revised version.

Point 6: "Hemicellulose (0.5 g) was added to 25 mL of pure water. The mixture was stirred at 60°C until it was completely dissolved. Initiator (NH4)2S2O8, acrylamide and cross-linking agent N, N-methylene diacrylamide were added in turn and stirred to mix evenly. " (Lines 95-97) - what are the exact amounts? (Section 2.4.)

Response 6: Amounts of various chemicals were added in revised version.

Point 7: Please correct these mistakes and other shortcomings of the experimental section, which I did not list.

Response 7: The mistakes and other shortcomings of the experimental section were modified.

Point 8: "The hydrolytic solution after membrane separation was neutralized to neutral." (Line 79) - neutralized means already that the pH is neutral

Response 8: The inappropriate expression has been modified.

Point 9: Headlines should not be separated from the sections they describe (Line 129)

Response 9: It has been modified according to your suggestion.

Point 10: Insets in Fig. 2a, 3a, 4a are too small to read. The same is valid for Fig. 5c. Same micrographs should be enlarged and have professional scale bar markers. Currently, no conclusions can be drawn from these images as they are too tiny.

Response 10: All Figures are modified and rearranged.

Point 11: "The nitrogen contents in the modified hemicellulose samples were analyzed by an elemental analyzer and found to be 0.10%, 0.23%, 0.45%, 0.55% and 0.80%." (Lines 84-86) and "Their dimensions were analyzed by transmission electron microscopy (TEM) and found to be 50 nm, 100 nm, 150 nm, 200 nm and 250 nm. "(Lines 91-92). It is puzzling how the authors obtained such round values. Please comment on that. Also, provide standard deviations for these values as there was definitely some uncertainty. I also invite the authors to include these TEM images in the final work.

Response 11: The above data are averages of multiple measurements. The standard deviations and uncertainty for these values was added. These TEM photographs were added in the attachment supporting materials.

Point 12: The conclusions section should be extended to include the impact of this contribution and future outlook. 

Response 12: It has been modified according to your suggestion. The impact of this contribution and future outlook was added in revised version.

As a whole, issues the referee suggested are very pertinent, which are very helpful to modify my entire paper and thank you very much again.

Round 2

Reviewer 2 Report

In general, I am pleased with the authors responses to my concerns, and the general improvements made to the manuscript after the initial assessment. I am happy to recommend the manuscript for publication in the current form.

Author Response

Thank you for your valuable comments.

Reviewer 3 Report

Thank you for following the suggestions. The article may be accepted in the present form. 

Author Response

Thank you for your valuable comments.